# Control over the fibrillization yield by varying the oligomeric nucleation propensities of self-assembling peptides

Chun Yin Jerry Lau [1], Federico Fontana[2], Laurens D. B. Mandemaker[4], Dennie Wezendonk[4], Benjamin Vermeer[5], Alexandre M. J. J. Bonvin[5], Renko de Vries[6], Heyang Zhang[7], Katrien Remaut[7], Joep van den Dikkenberg[1], João Medeiros-Silva [5], Alia Hassan[8], Barbara Perrone[8], Rainer Kuemmerle[8], Fabrizio Gelain [2,3], Wim E. Hennink[1], Markus Weingarth [5,9✉] & Enrico Mastrobattista [1,9✉]

Self-assembling peptides are an exemplary class of supramolecular biomaterials of broad biomedical utility. Mechanistic studies on the peptide self-assembly demonstrated the importance of the oligomeric intermediates towards the properties of the supramolecular biomaterials being formed. In this study, we demonstrate how the overall yield of the supramolecular assemblies are moderated through subtle molecular changes in the peptide monomers. This strategy is exemplified with a set of surfactant-like peptides (SLPs) with different β-sheet propensities and charged residues flanking the aggregation domains. By integrating different techniques, we show that these molecular changes can alter both the nucleation propensity of the oligomeric intermediates and the thermodynamic stability of the fibril structures. We demonstrate that the amount of assembled nanofibers are critically defined by the oligomeric nucleation propensities. Our findings offer guidance on designing self-assembling peptides for different biomedical applications, as well as insights into the role of protein gatekeeper sequences in preventing amyloidosis.

[1] Utrecht Institute for Pharmaceutical Sciences, Department of Pharmaceutics, Faculty of Science, Utrecht University, Universiteitsweg 99, 3584 CG Utrecht, The Netherlands. [2] IRCCS Casa Sollievo della Sofferenza, Opera di San Pio da Pietralcina, Viale Capuccini 1, 71013 San Giovanni Rotondo, Italy. [3] ASST Grande Ospedale Metropolitano Niguarda, Center for Nanomedicine and Tissue Engineering, Piazza dell'Ospedale Maggiore 3, 20162 Milan, Italy. [4] Inorganic Chemistry and Catalysis Group, Debye Institute for Nanomaterials Science, Department of Chemistry, Faculty of Science, Utrecht University, Universiteitsweg 99, 3584 CG Utrecht, The Netherlands. [5] NMR Spectroscopy, Bijvoet Centre for Biomolecular Research, Department of Chemistry, Faculty of Science, Utrecht University, Padualaan 8, 3584 CH Utrecht, The Netherlands. [6] Laboratory of Physical Chemistry and Colloid Science, Wageningen University, Dreijenplein 6, 6703 HB Wageningen, The Netherlands. [7] Ghent Research Group on Nanomedicines, Laboratory of General Biochemistry and Physical Pharmacy, Ghent University, Ottergemsesteenweg 460, 9000 Ghent, Belgium. [8] Bruker BioSpin AG, Industriestrasse 26, 8117 Fällanden, Switzerland. [9] These authors jointly supervised this work: Markus Weingarth, Enrico Mastrobattista. ✉email: m.h.weingarth@uu.nl; e.mastrobattista@uu.nl

Self-assembling peptides, a prominent class of supramolecular polymers, can form well-ordered nanostructures via non-covalent interactions (i.e., van der Waal's forces, electrostatic forces, hydrogen bonding) as the main modulators for material tailoring[1]. Due to the dynamic and reversible nature of their interactions, self-assembling peptides offer novel functional properties for practical exploitation, e.g., multicomponent modularity[2–4], semiconductivity[5,6], and evolution-like adaptivity[7].

In search for the link between the peptide monomers and the final assembly states, a number of previous reports on self-assembling peptides have studied the effect of changing the molecular properties of peptide monomer (e.g., sequences' residues or stereochemistry[8,9]) toward the final assembled products. Many studies have treated the peptide self-assembly as a spontaneous thermodynamic process[8–10]. Therefore, the linear correlation between the properties of peptide monomers and the final assembled structures are often described in these studies. However, increasing evidence shows that, to overcome the huge desolvation barrier, rather than a spontaneous thermodynamic process, supramolecular assembly of amphiphilic peptides proceed via a multistep[11] pathway, along which metastable oligomeric states are first formed before conversion to supramolecular nanofibers (Fig. 1a)[12–18]. This implies that the state of the intermediates in the assembly pathway also exert critical influence over the outcome of the peptide self-assembly[19,20]. For example, the polymorphic form of the assembled peptide fibrils are influenced by the properties of the oligomeric intermediates[21]. However, despite progress in the mechanistic understanding, the interrelationship between the molecular properties of the peptide monomers, the oligomeric intermediates and the overall yield of the supramolecular assembly process remains largely elusive[22].

Here we used surfactant-like peptides (SLPs) to show that the yield of peptide fibrillization are controlled by the properties of the oligomeric intermediates, which can be moderated by subtle molecular variations in the peptide sequences. SLPs consist of two modular subunits (Fig. 1b): hydrophobic tails that interact via van der Waals' forces (side chain) and hydrogen bonds (backbone), as well as hydrophilic headgroups that confer mutually repulsive electrostatic interactions and determine the overall charges of the peptides[23]. The modularity of SLPs allow us to single out one molecular property at a time and study the effect it exerts in the downstream self-assembly pathway. We composed a set of cognate SLPs with small molecular variations (Fig. 1b). Using combined experimental techniques and molecular dynamics (MD) simulations, we demonstrate that these two molecular parameters (i.e., β-sheet propensity and charged residues flanking the aggregation domains) can modulate both the nucleation propensity of the oligomeric intermediates and the thermodynamic stability of the fibril structures. We demonstrate that the amount of peptide nanofibers being formed are critically defined by the oligomeric nucleation propensities. Altogether, our results offer a general molecular approach to moderate the properties of peptide assemblies for a variety of biomedical applications.

## Results and discussion

**Molecular design of SLPs**. First, we designed a set of four cognate SLPs with different conformational propensities and headgroup charges in order to probe the effect of these molecular differences on the assembly pathway. We chose branched-chain amino acids (valine, leucine, isoleucine) as the major building blocks for the hydrophobic tail subunit, which are residues that often confer structural stability in proteins[24]. To limit the amount of molecular variability, we used Leu/Ile residue isomerism as the strategy for specific β-sheet propensity variation[25]. Ile has a higher β-sheet propensity compared to Leu but substitution does not change the overall side-chain molecular volume and hydrophobicity. Since the sequence order can also influence the overall β-sheet propensity[26], we employed the position scoring matrix WALTZ[27] to guide sequence design. The length of the tail group was chosen as

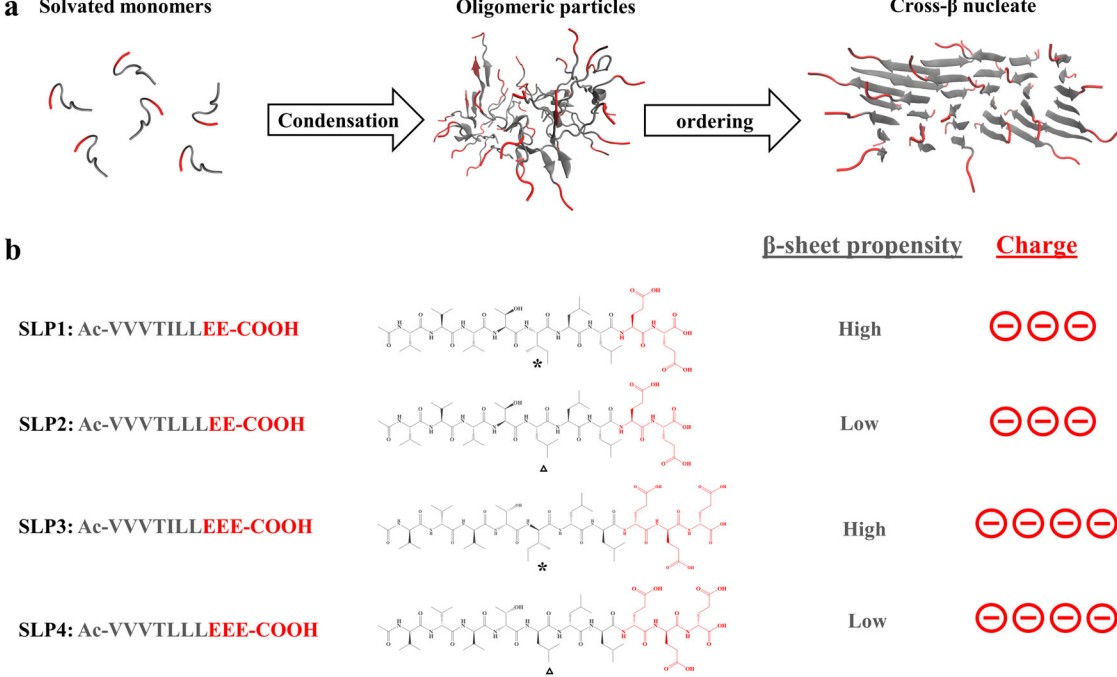

**Fig. 1 Modularly engineered surfactant-like peptides (SLPs). a** Schematic representation of the two-step nucleation mechanism of peptide self-assembly, in which peptides first assemble into oligomeric particles through condensation; nucleates are then formed within the oligomeric particles. **b** Primary sequence and chemical structures of the studied SLPs. Hydrophobic tails and anionic headgroups are colored in gray and red, respectively. The mutated amino acids are highlighted by an asterisk for Ile and a triangle for Leu.

N-terminal acetylated 7-mer peptide. An extra amino acid and an N-terminal acetyl cap was added to the 6-mer peptide sequence to minimize the influence of flanking effects by the charged residues[28]. The WALTZ database suggests that position 5 is a highly selective position for isoleucine, but not leucine, to drive cross-β structure formation. We also placed threonine between two aliphatic amino acid trimers in order to create two β-sheet faces of different hydrophobicity. This allowed assignment of the anisotropic side-chain interface that fits with the statistical mechanical model of fibril assembly (Fig. S5)[29,30]. The statistical thermodynamics algorithm TANGO was used to determine the residual aggregation propensities along the whole sequence (Table S1)[31]. Moreover, the number of charges in SLP was adjusted by altering the number of glutamic acid residues in the headgroup (Fig. 1b).

**Headgroup charges regulate the size distribution of the oligomeric particles**. To study the effect of headgroup charges toward the size of the metastable oligomeric particles, we employed multiscale MD simulations to confirm the structural arrangement of the oligomeric particles (see "Methods"). The final trajectory of atomistic simulations suggested that the SLP oligomeric particles adopt a micellar arrangement with surface-exposed headgroups and buried tail groups (Figs. 2a and S1 and S2). Next, given that higher surface electrostatic repulsive forces can impose higher repulsive forces between the oligomeric particles, thereby lower their coalescence tendency[32], we speculated

that the size of the micellar oligomer with three glutamic acid residues in the headgroup (SLP3–4) should be smaller than that with two glutamic acids (SLP1–2). To validate this hypothesis, we performed fluorescence correlation spectroscopy (FCS) measurements to study the diffusion properties of SLPs in the early phase of self-assembly. FCS was chosen to inspect the early assembly phase, as it gives high-resolution measurement for the small-sized oligomeric particles in solution state[33]. By comparing the autocorrelation graphs of SLP1–4 (Fig. 2b), we could confirm that the size of oligomers decreases with the number of negative charges in the peptide headgroup, i.e., SLPs with two negative charges (SLP1–2) form larger oligomers than SLPs those with three negative charges (SLP3–4). To explore these size differences in more detail, we fitted the FCS data with a higher-order fitting model. Maximum entropy method (MEM) was chosen to account for the polydispersity of the oligomers. Each condensed fraction was treated as one quasi-continuous distribution, so as to provide the widest data-complying size distribution (least chance of overinterpretation)[34]. We succeeded to resolve the monomeric peptides and oligomers for SLP1–3 (Fig. 2c), whereas, due to their close size range, monomers and oligomers were represented as one continuous distribution for SLP4 (Fig. 2c). In the MEM analysis, the diffusion coefficients of the SLP1–2 oligomers were ~10-fold smaller than for SLP3–4, clearly showing a correlation between headgroup charge and the size of oligomers, i.e., oligomers with two glutamates in the headgroup are considerably

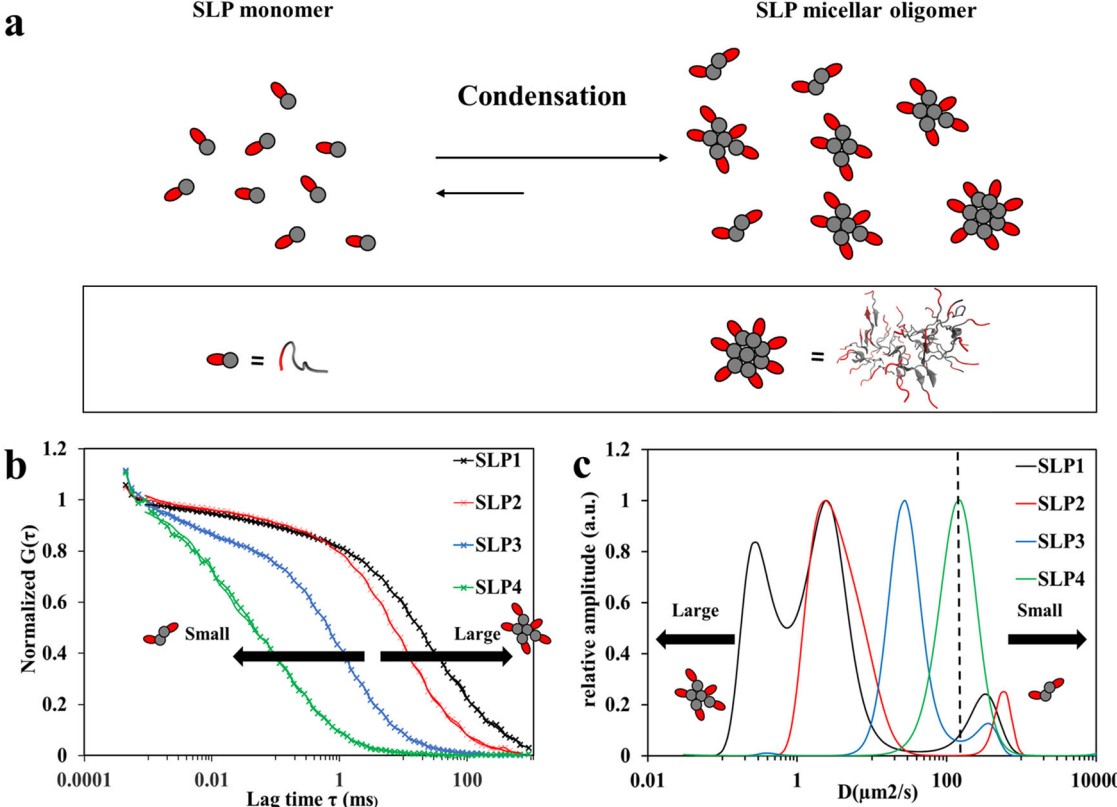

**Fig. 2 Effect of headgroup charge on SLP oligomerization. a** Illustration of the LLPS mechanism for SLPs. Micellar arrangement of the oligomers is caused by the amphiphilic nature of SLP[59]. Representative structures for monomer and micellar oligomers derived from MD simulations are shown below. **b** Normalized autocorrelation curves of SLP1–4 determined by fluorescence correlation spectroscopy (FCS) as performed for Cy5-labeled SLPs (black, red, blue, and green crosses). FCS data show that the assemblies of SLP1–2 are globally larger than that of SLP3–4. The measurement concentration was 4 mM of SLPs in PBS (pH 7.4) of which 1 out of 4000 peptides was labeled with Cy5. The FCS autocorrelation curves were fitted with the maximum entropy method (MEM) for higher-resolution analysis on size distribution, indicated by the solid lines. **c** The size distribution of SLPs obtained from MEM analysis of the FCS measurements. The dashed line at $D = 150\ \mu m^2/s$ indicates the cut-off size between monomeric ($D > 150$) and oligomer ($D < 150$) populations. SLP1–2 with two glutamic acid residues in the headgroup formed oligomers with ~10 times slower diffusivity than SLP3–4 with three glutamates in the headgroup.

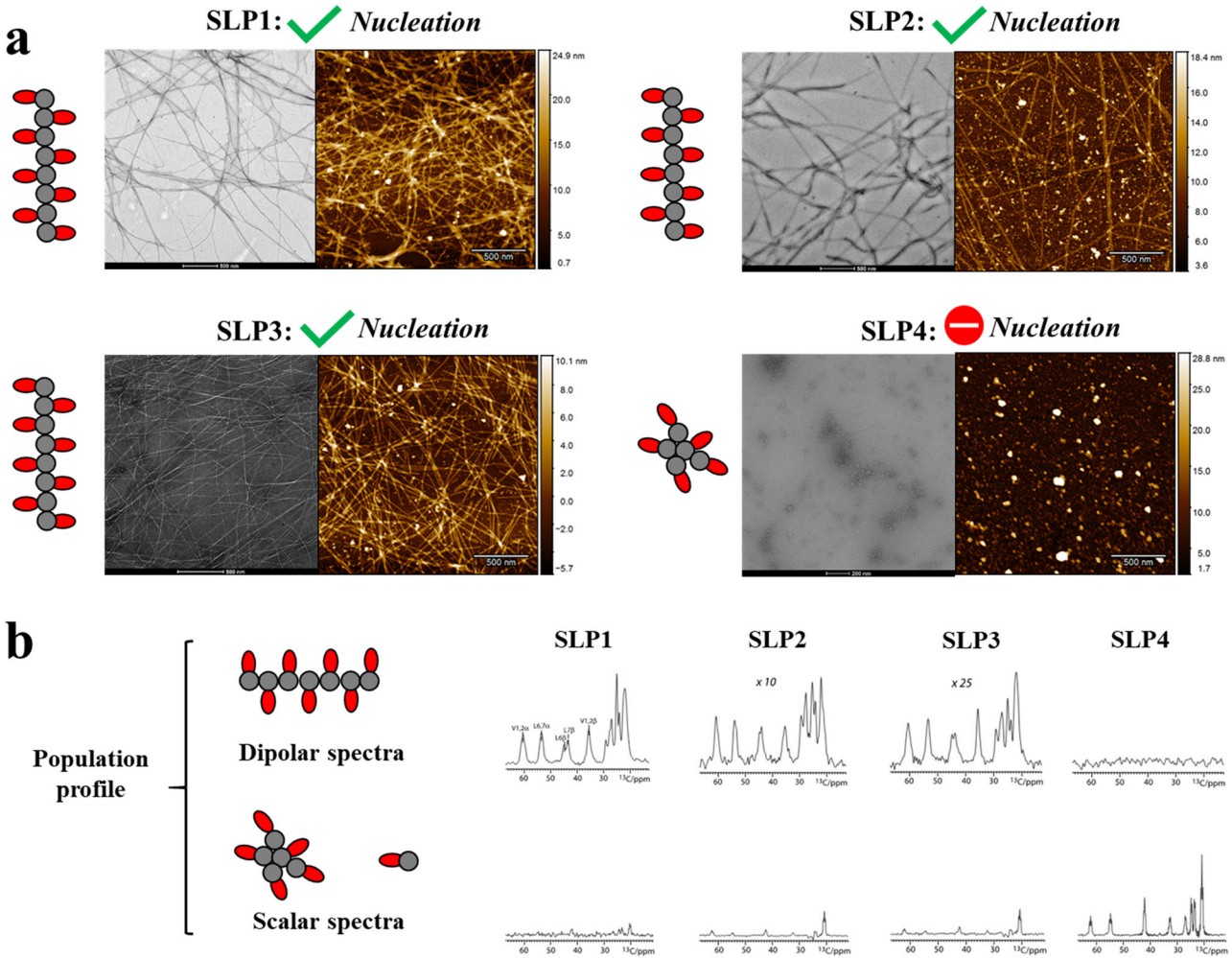

**Fig. 3 Microscopic images of SLP assemblies and solid-state NMR-derived population profiles. a** Negative staining TEM (left panel) and AFM images (right panel) showing the assembled structures of SLP1–4. 1D nanofibers are formed with SLP1–3, indicating that nucleation event have taken place. Only 0D nanostructures are observed in SLP4, indicating that no nucleation event has taken place. **b** Population profiling of SLP assemblies by solid-state NMR spectroscopy. The dipolar cross-polarization (CP) spectra report on rigid 1D nanofibers and the scalar INEPT spectra report on mobile species (micellar oligomers and monomeric SLP). The dipolar spectra were normalized (scaling factors of 10 and 25 for SLP2 and SLP3, respectively), while the intensity in scalar spectra directly reflects on the mobile population in the system. The spectra were measured at 500 MHz ($^1$H frequency), 10 kHz magic angle spinning (MAS), and at 280 K sample temperature.

larger than those with three glutamates. Recent studies show that nucleation events only happens in a minority of the oligomer populations[13,35]. Therefore, although this analysis does not allow us to distinguish between larger oligomers and the early nucleated structures (particularly the 0.1–1 µm²/s SLP1 population in Fig. 2c), as they fall in the same size range, we were able to validate our hypothesis that headgroup charges determine the size distribution of the oligomeric particles.

**Mesoscale structure of SLP assemblies characterized by microscopy.** Next, we investigated the downstream effect of the differences in size of oligomeric particles and β-sheet propensities on the supramolecular self-assembly. One of the pivotal events in defining supramolecular self-assembly is whether nucleation has taken place or not. This can be identified by the mesoscale structures of the assemblies after certain time of incubation, e.g., the formation of one-dimensional (1D) nanostructures is indicative that nucleation has taken place. We used negative-staining transmission electron microscopy (TEM) and atomic force microscopy (AFM) to compare SLPs that were assembled under the same preparation protocol (4 mM, pH 7.4 in phosphate-

buffered saline (PBS) for 3 days). We could indeed detect markedly different mesoscale structures in these SLP systems (Figs. 3a and S1). While SLP1–3 formed 1D nanofibers (~10 nm in diameter, >1000 nm long), we observed polydisperse zero-dimensional nanostructures for SLP4 (diameter of 10–60 nm). Based on these mesoscale evidences, we can deduce that nucleation occurred in SLP1–3, however, not in SLP4.

**Determination of population size of fibril and non-fibril assemblies by solid-state nuclear magnetic resonance (ssNMR) spectroscopy.** As a next step, we hypothesized that higher-resolution methods might disclose further differences in the properties of the SLP assemblies. To examine our hypothesis, we profiled the SLP1–4 assemblies using ssNMR spectroscopy. Using the same conditions (4 mM, pH 7.4 in PBS for 3 days), we acquired the so-called dipolar cross-polarization (CP) ssNMR spectra and so-called scalar INEPT ssNMR spectra on isotopically ($^{13}$C, $^{15}$N) labeled SLP assemblies (see "Methods"), which enables to quantify rigid and mobile populations of peptide assemblies in the systems[36] (Fig. 3b). While dipolar signals report on immobilized peptides in 1D assemblies, scalar spectra report on

peptides with fast pico-to-nanosecond dynamics, which can either be monomeric peptides or micellar oligomers[16,32]. We specifically labeled residues in the hydrophobic tail (Val1–2 and Leu6–7) that are responsible for the self-assembly. In line with the mesoscale differences observed by microscopy (Fig. 3a), ssNMR showed stark differences in the population profiles of SLP1–3 and SLP4 (Fig. 3b). SLP1–3 showed sizeable dipolar signals, indicating that a considerable number of peptides formed immobile assemblies, whereas dipolar signals were absent for SLP4, in line with the absence of 1D nanofibers for SLP4. A closer inspection into the population profiles of SLP1–3 revealed noticeable difference in the dipolar and scalar signals between SLP1–3. The intensity of the dipolar spectra correlates with the amount of rigid cross-β structure present in the system. We observe that the intensities of dipolar signal of SLP1 is larger than SLP2 (normalization scaling factor of 10, Fig. 3b) and SLP3 (normalization scaling factor of 25, Fig. 3b), which indicate the relative fibrillization yield of SLP1 > SLP2 > SLP3. Besides, the scalar signals were higher for SLP2–3 than for SLP1, which means that SLP2–3 have a higher number of mobile peptides than SLP1.

Furthermore, ssNMR signals are sensitive reporters of the peptide conformation in the assemblies[28,36]. We assigned [15]N and [13]C signals and amino protons in the assembled SLP1 system using two-dimensional (2D) [13]C–[13]C PARIS[37], 2D CαN, and [1]H-detected 2D Cα(N)H experiments in combination with peptides in which only residues Val2 and Leu6 were isotope labeled. These assignments unambiguously show that assembled peptides adopt β-strand configuration, while the mobile population in the system is unstructured (Fig. 4c).

**Reconstruction of fibril models**. Next, we sought to elucidate how these molecular variations can affect the thermodynamic properties of the fibril assemblies. Inspired by a previous computational approach[38], we built models of the SLP fibrils and validated them with the help of X-ray diffraction (XRD) and ssNMR spectroscopy. SLP1–3 fibers exhibit typical cross-β XRD pattern, showing reflections at ~4.7 and 9–11 Å (Fig. 5c). Since XRD (sharp reflection at 4.7 Å in XRD) and ssNMR show that SLP1–3 share the same interstrand features (Fig. 3b), we used SLP1 as representative to elucidate the inter-β-strand configuration. We acquired a 2D [13]C–[13]C PARIS ssNMR spectrum with a long magnetization transfer time of 700 ms that probes distances between [13]C nuclei with a threshold of ~8 Å. Given that the Cα$_i$–Cα$_{i+5}$ distance within the same β-strand is 15–18 Å and hence markedly beyond the ssNMR distance threshold, our labeling scheme can conclusively distinguish parallel or anti-parallel alignment of β-strands (Fig. 4b). However, measuring intermolecular contacts between the peptides in our dilute experimental concentration (4 mM) is a serious sensitivity challenge, and increasing the sample concentration was not possible because it could alter the kinetic pathway of fibril assembly. To this end, we used a CPMAS CryoProbe prototype (BioSolids CryoProbe[TM], Bruker Biospin)[39] that markedly enhanced spectral NMR sensitivity. With this advanced experimental set-up, we were able to observe a large number (>50) of intermolecular peptide–peptide contacts and could unambiguously establish an antiparallel alignment of β-strand in the fibers (Fig. 4a, b). Interestingly, while Leu6 showed intermolecular NMR correlations with all aliphatic carbons of Val1 and Val2, some intermolecular correlations were absent for Leu7. This suggests that the β-strand tails form interdigitated antiparallel arrangements from which the charged headgroups stick out in order to minimize electrostatic repulsion.[28] The antiparallel, interdigitated dimer configuration was confirmed by NMR structure determination (Fig. 4d) for which we used NMR distance restraints and

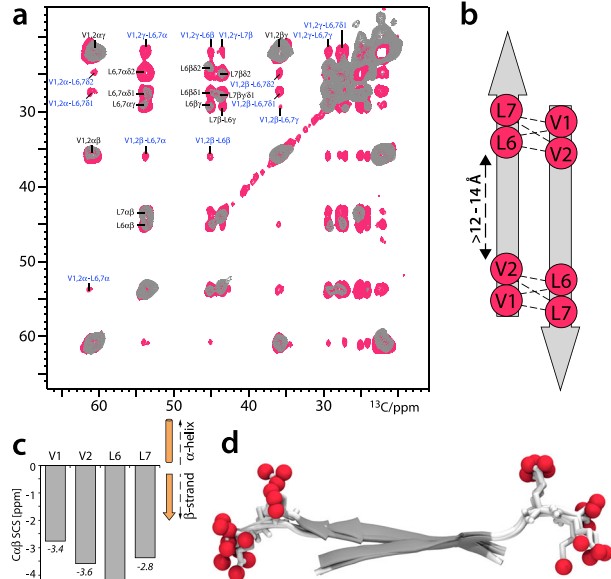

**Fig. 4 Structure determination of the basic building blocks of the SLP fibers. a** Superposition of 2D ssNMR PARIS [13]C–[13]C spectra of [13]C,[15]N-(Val1, Val2, Leu6, Leu7)-labeled SLP1 acquired with 50 (gray) and 700 ms (magenta) magnetization transfer. Intramolecular and intermolecular correlations are labeled in black and blue, respectively. The spectrum with 700 ms magnetization transfer time was acquired with a CPMAS CryoProbe (Bruker Biospin). **b** Inter-residual magnetization transfer from Val1/Val2 to Leu6/Leu7 observed in the 2D [13]C–[13]C ssNMR spectrum relates to intermolecular contacts between antiparallel β-strands. Distances are Cα–Cα spacings. **c** ssNMR CαCβ secondary chemical shifts show that the hydrophobic tails adopt β-strand conformation in the SLP1 nanofiber, which is similar for SLP2 and SLP3 fibers[60]. **d** ssNMR structure of the SLP1 dimer in the 1D nanofiber. A superposition of the three best structures is shown (backbone RMSD of 1.11 Å). Oxygen atoms of the anionic C-termini are highlighted as red spheres.

dihedral restraints[40]. The antiparallel alignment was also in line with [15]N R1$_{rho}$ measurements that probe the slow microsecond dynamics of the assembly (Fig. S4). While the β-strand residues Val1–Leu7 are generally highly rigid, residue Leu7 showed modestly enhanced dynamics, in line with charge-flanking residue effect observed in antiparallel β-sheet arrangement previously[28].

Next, we built structural models of the SLP fibrils. Therefore, based on the established interstrand configuration, we arranged side-chain faces of different hydrophobicity following the blue-print outlined from previous statistical mechanical model[29,30] (Fig. S5) and built MD fibril models composed of 36 peptides for each of SLP1–3 (Figs. 5a and S6). Each fibril model was simulated for 100 ns. Fibrils remained stably associated over the entire trajectory. In agreement with previous reports, a left-handed twist was observed for all three fibril models due to the chirality of the constituting L-amino acids (Figs. 5b and S6)[29,30]. Radial distribution function (RDF) of the inter-backbone distances were calculated from the fibril model and cross-validated with XRD results. The RDF and XRD results are in mutual agreement, showing signal peak at ~4.7 and 9–11 Å (Fig. 5c, d). The good match between several experimental results and our MD model strongly corroborates the validity of our fibril models.

**Thermodynamic stability assessment by steered MD (SMD) simulations**. As a next step, we evaluated the thermodynamic stability of the fibril models using SMD simulations. Analogously,

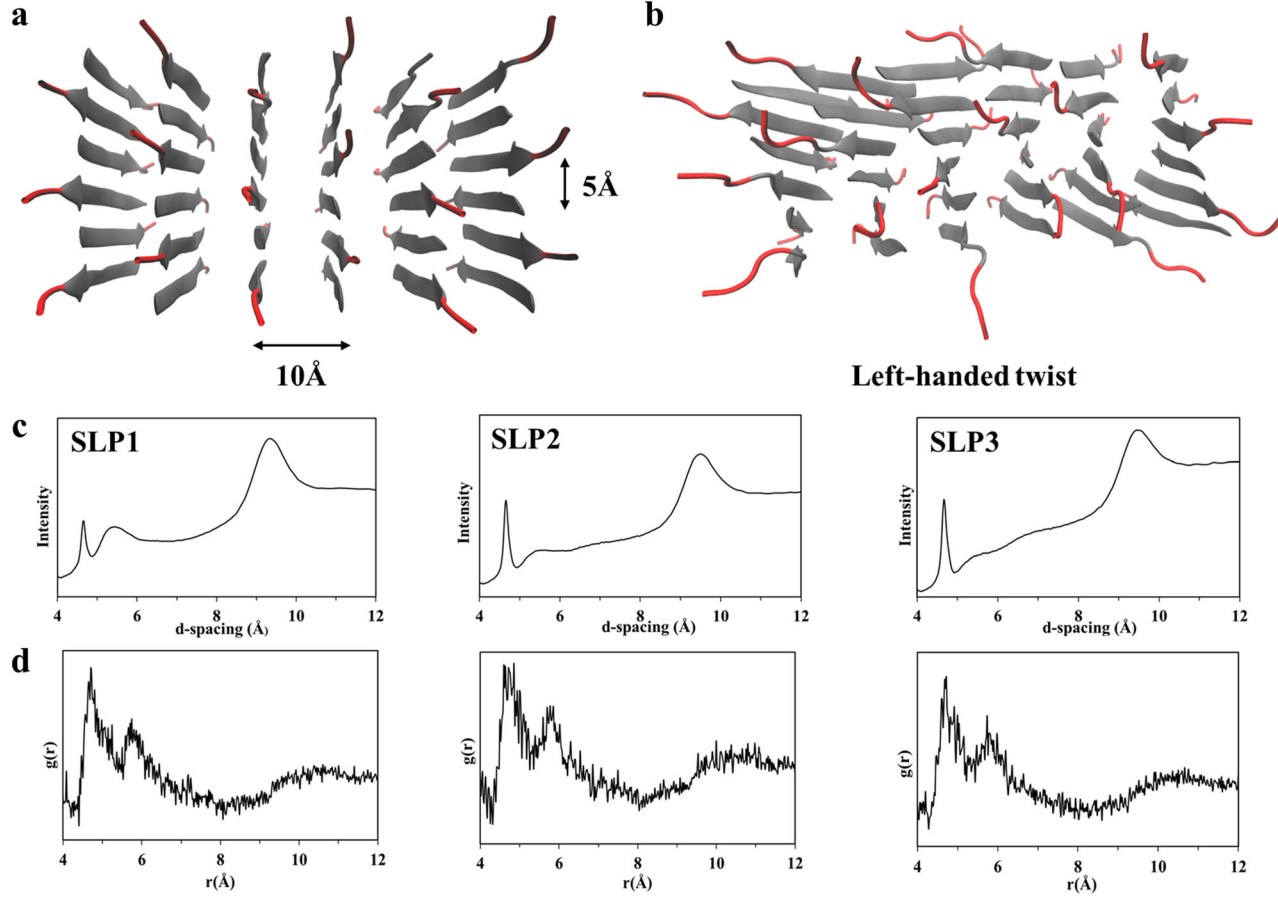

**Fig. 5 Fibril models derived from MD simulations. a** Representative atomistic manually built model of the starting fibril configuration with antiparallel SLPs. SLPs within a β-sheet are 5 Å apart, and 10 Å is the orthogonal distance between SLPs. **b** Representative final configuration of the equilibrated molecular models after 100 ns of MD simulation. The chirality of the L-amino acids leads to left-handed twist as is observed in the SLP models. **c** XRD pattern of the SLP1–3 fibrils. The reflection at ~4.7 and 9–11 Å represent the signatory of cross-β diffraction pattern. **d** Radial distribution function (RDF) of the backbone–backbone distance calculated from the final MD configuration of SLP fibrils. In agreement with XRD, RDF also show peaks at ~4.7 and 9–11 Å, demonstrating a good match between MD fibril models and experimental data.

we also determined the thermodynamic stability of the oligomer structures. In these pulling simulations, an external mechanical force is applied to one SLP, which is then dragged from an aggregate core. With this approach, we derived a potential of mean force (PMF) profile, which is a good representation of the dissociation energy (fibril: ΔGd, oligomer: ΔGd')[41,42]. Since the choice of the pull-out SLP within the molecular model determines the resultant PMF profiles, we chose one random coil-forming SLP and one β-sheet SLP (Figs. 6a and S7). Each pull-out SLP was dragged along the *x*-direction (the reaction coordinate *r*) for 90 Å, while the other 35 SLPs were constrained by the application of a harmonic force. These results show that it takes ~3-fold more energy to pull one SLP out from the fibril structure than from the oligomer structure (Figs. 6a and S8). This means that fibril structures are much more stable configurations than the oligomer. For the fibril models, the increased headgroup charge in SLP3 caused reduction in ΔGd and less fibril stability compared to SLP1–2. For the oligomer models, however, the charge differences in the peptide headgroups did not cause large differences in the ΔGd'. Since the magnitude of the dissociation energy is an indicator for the thermodynamic stability of the supramolecular assemblies, these results suggested that an increasing number of charges in the headgroup caused more perturbations to the more ordered fibril structures than to the less ordered oligomer. We suspected that these differences are due to the structural elasticity. The structural strain caused by increased electrostatic repulsion

forces would be more detrimental for brittle rigid structures (fibrils) than for flexible structures (oligomers). Such differences in structural elasticity are well supported by the strongly deviating sizes of the rigid and mobile equilibrium populations that we observed for SLP1–3 and SLP4 with the ssNMR experiments described above.

**The yield of nanofibers is moderated by the nucleation propensity in the oligomeric intermediates.** After characterizing the properties of SLP at the oligomer and fibril states, we sought to search for the underpinning reason behind the differences in yield of peptide nanofibers between SLP1–4. It was suggested that the final yield of the peptide nanofibers is determined by its structural stability[10]. However, with the high thermodynamic stability of the peptide nanofibers, the amount of monomeric peptide dissociation from fibrils are typically negligible[43]. Indeed, we have observed that SLP1 and SLP2, although their fibril exhibits similar thermodynamic stability, they show markedly different amounts of fibril and non-fibril assemblies. We therefore rationalize the difference in yields of nanofibers between SLP1–4 is instead influenced by the properties of the oligomer intermediates. Due to the heterogenic nature of the oligomeric species, it was reported that nucleation only happens in a minority of the oligomers[13]. Furthermore, the nucleation probability is correlated with the size of the oligomers, in which the chance of nucleation increases with

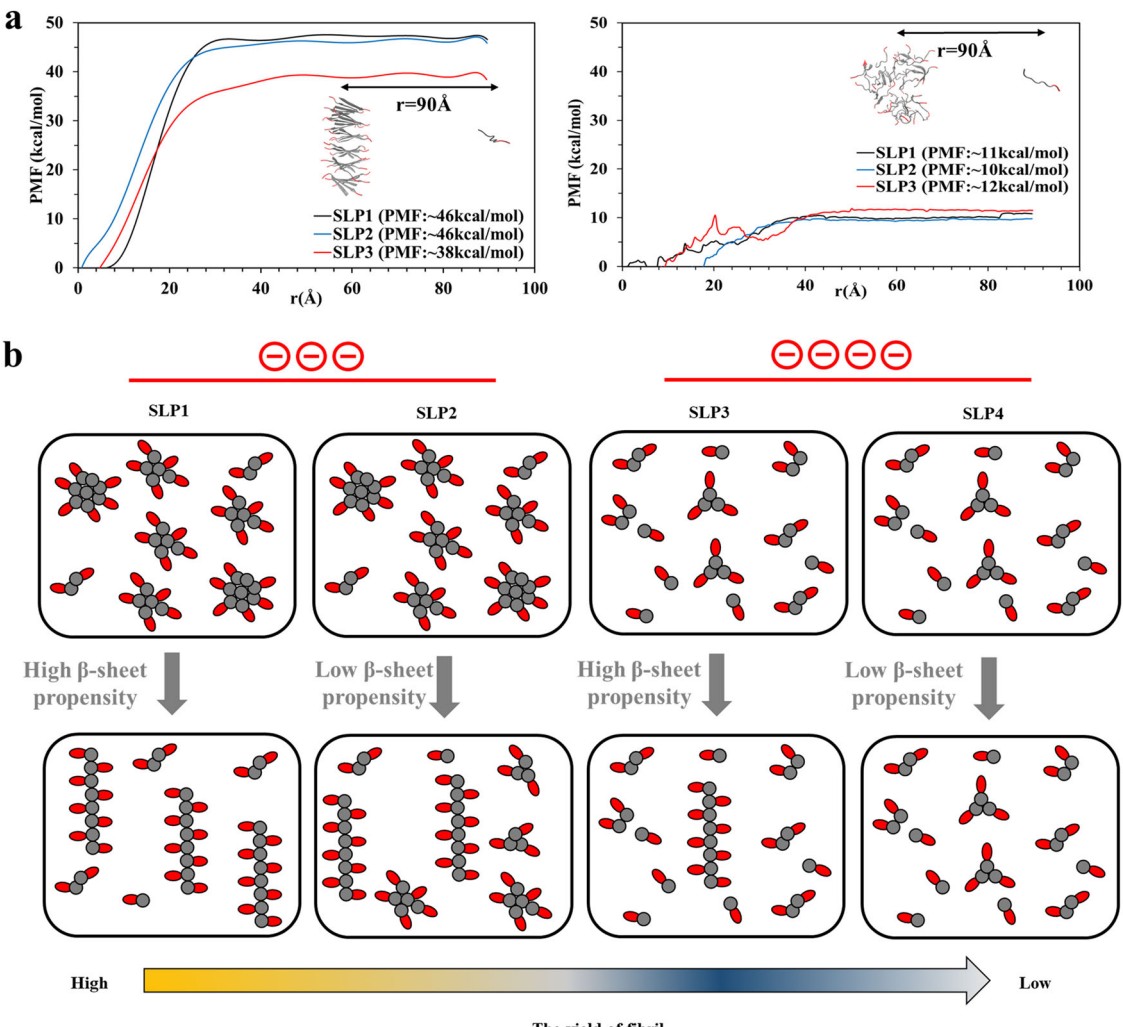

**Fig. 6 Control over the fibrillization yield by varying the oligomeric nucleation propensies of surfactant-like peptides. a** Potential of mean force (PMF) profiles of fibril (left) and oligomer (right) models along the reaction coordinate *r* derived from steered MD (SMD) simulations provides information on the thermodynamic stability of fibril and oligomer structures. Increasing the number of charges of the headgroups reduced structural stability of the fibrils, however, not in the oligomers. A representative SMD trajectory, during which an SLP monomer is dragged from the core of oligomer and fibril model for 90 Å along *r*, is shown under each graph. **b** Graphical representation of the proposed mechanism leading to the different yields of nanofibers between SLP1 and SLP4. The size distribution of the oligomeric intermediates is controlled by the number of headgroup charges, which the less charged SLPs form bigger-sized oligomer (two on the left) and the more charged SLPs form smaller-sized oligomers (two on the right). Between the SLPs with the same charges, the fibrillization propensity is modulated by the β-sheet propensities of the tail group, with the SLP of higher β-sheet propensities giving higher yields of fibrils, i.e., SLP1 > 2 and SLP3 > 4.

the oligomer size[16,35]. Indeed, we observed that SLP1–2, which form larger-sized oligomers than SLP3–4, also gives higher nanofiber yields (Fig. 3b). Besides, the nucleation probability is also influenced by the conformation propensity of the SLPs, in which more β-sheet-prone SLPs have a higher chance of forming cross-β nucleates. Therefore, for the SLPs that bear the same headgroup charges, we consistently observed higher yield of nanofibers for those with more β-sheet-prone tail groups, i.e., SLP1 > 2 and SLP3 > 4 (Fig. 3b).

Our findings advance our fundamental understanding of the molecular design principles of SLPs (and self-assembling peptides in general) to tailor their properties for various applications (Fig. S9). For applications that require that SLPs stay in dynamic form, e.g., the solubilization of membrane proteins[44], one should increase the charges and lower the β-sheet propensities of SLPs in order to prevent fibrillization, as we here demonstrate it for SLP4. In contrast, for applications that require the fibril infrastructure, e.g., for therapeutic scaffolds[2–4], one should engineer the SLP

with higher β-sheet propensities and lower number of charges to maximize the yield of fibril structures (like SLP1). In addition, for applications that use 1D nanofibers as reservoir for hydrophobic drug release[45], the amount of micellar oligomer co-present in the system can affect the overall drug release profile. At last, with respect to protein aggregation, a previous genome-wide sequence analysis revealed that, in close proximity to the aggregation-prone β-strand regions in proteins, charged "gatekeeper" residues often prevent the β-strand sequences from aggregating[46]. Our current study implies that, on top of preventing aggregation, the charged protein gatekeeper sequences can also lower the chance of amyloid nucleation through lowering the size of the oligomers.

## Conclusion
The data presented here demonstrated, for the first time, the molecular strategy to modulate the yield of the supramolecular polymerization process. The effect of the molecular changes

toward the size distribution of the oligomeric particles and the thermodynamic stability of the fibrils are also evaluated, which helps defining the mechanism leading to the differential polymerization yield. Thereby, our study enhances our understanding of the important role of oligomeric intermediates in defining the outcome of self-assembly systems and advances the rational design principles of self-assembling peptides that give different supramolecular properties for a wide latitude of applications[2–4,44,45]. Moreover, our findings suggest that charged protein gatekeeper sequences[46] can prevent amyloidosis by lowering the size of the oligomeric particles. This knowledge will be instrumental for the design of strategies to prevent amyloidosis.

## Methods

**Materials**. Preloaded Fmoc-Glu(Otbu)-Wang resin was purchased from Novabiochem GmbH (Hohenbrunn, Germany), 2-(1H-Benzotriazol-1-yl)-1,1,3,3-tetra-methyluronium hexafluorophosphate (HBTU), 9-fluorenylmethyloxycarbonyl (Fmoc)-protected amino acids, and trifluoroacetic acid (TFA) were purchased from Iris Biotech (Marktredwitz, Germany). Peptide-grade *N*-methyl-2-pyrrolidone (NMP), dichloromethane, piperidine, *N,N*-diisopropylethylamine (DIPEA), and high-performance liquid chromatography (HPLC)-grade acetonitrile (ACN) were purchased from Biosolve BV (Valkenswaard, Netherlands). 1-Hydroxy-benzotriazole hydrate (HOBt), triisopropylsilane (TIPS), and BioUltra-grade ammonium bicarbonate and sodium bicarbonate were purchased from Sigma-Aldrich Chemie BV (Zwijndrecht, Netherlands). $^{13}$C,$^{15}$N-labeled Fmoc-amino acids were purchased from Cortec-net (Les Ulis, France).

**Bioinformatic analysis**. To predict the aggregation and fibrillization propensity of the designer SLPs, statistical thermodynamics algorithm, TANGO[31], and position scoring matrices WALTZ[27] were used to calculate the respective scores (available at http://tango.crg.es/tango.jsp and http://waltz.switchlab.org/).

**Solid-phase peptide synthesis and characterization**. The SLP were synthesized by standard Fmoc solid-phase peptide synthesis using Symphony peptide synthesizer (Protein Technologies, US). NMP was used as the coupling and washing solvent for the whole synthesis process. For each coupling step, Fmoc-amino acids were activated by 4 eq HBTU/HOBt and 8 eq DIPEA to react with the free N-terminal amino acids in the resin for 1 h. After each coupling step, Fmoc group was removed by fourfold treatment of 20% piperidine for 10 min. TFA/water/TIPS (95/2.5/2.5) was used to simultaneously cleave the peptide off from the resin and remove the side-chain protecting groups. Peptides were purified by Prep-HPLC using Reprosil-Pur C18 column (10 μm, 250 × 22 mm) eluted with water–ACN gradient 5–80% ACN (10 mM ammonium bicarbonate) in 40 min at flow rate of 15.0 ml/min with ultraviolet (UV) detection at 220 nm. Purity was confirmed to be >90% by analytical RP-HPLC using Waters XBridge C18 column (5 μm, 4.6 × 150 mm) eluted with water–ACN gradient 10–80% ACN (10 mM ammonium bicarbonate) in 20 min at flow rate 1.0 ml/min and UV detection at 220 nm. Mass spectrometry (MS) analysis was performed using electrospray ionization (ESI)-LC/MS instrument (Supplementary Notes 1–8).

Peptides for Cy5 modification were synthesized as described above with addition of one lysine to the C-terminus. Peptides were then dissolved in 0.1 M sodium bicarbonate solution (pH 8.3) and Cy5 NHS ester (10 eq in 1/10 volume of dimethylformamide) was added and incubated overnight. Cy5-conjugated peptides were purified by Prep-HPLC. MS analysis was performed using ESI-MS instrument (Supplementary Notes 9–12).

**Sample preparation**. Peptide assemblies were prepared by dissolving peptide powders in nine volumes of 10 mM sodium hydroxide in 1.5 ml Eppendorf tube. One volume of PBS (10×) was added to make sample of pH 7.4 ± 0.2 and a final concentration of 4 mM. Solution was vortexed for 5 s and incubated statically for 3 days at room temperature before proceeding for measurements.

**X-ray diffraction**. XRD measurements were carried out on a Bruker-AXS D8 Advance powder X-ray diffractometer in Bragg–Brentano mode equipped with automatic divergence slit (0.6 mm 0.3°) and a PSD Våntec-1 detector. The radiation used was Co-Kα1,2, λ = 1.79026 Å, operated at 30 kV, 45 mA.

**Negative-staining TEM**. Samples prepared at 4 mM were diluted tenfold with 1× PBS. Formvar/carbon-coated 400 mesh copper grid (Polysciences Inc.) was placed on top of a droplet of 20 μl of diluted samples. After 2-min incubation, the grid was washed three times with 0.2 μM filtered mili-Q water and blotted dry with filter paper. Negative staining was performed for 1 min with 2% w/v uranyl acetate in water. Staining solution was blotted off with filter paper. Samples were imaged on a Tecnai 20 transmission electron microscope (FEI, Eindhoven, the Netherlands)

equipped with 4 K square pixel Eagle CCD camera (FEI, Eindhoven, the Netherlands) and operated at 200 kV accelerating voltage.

**AFM imaging**. AFM micrographs were recorded using a Bruker MultiMode 8 (ScanAsyst Air silicon nitride probes, spring constant 0.4 N/m, nominal tip radius 2 nm) and post-processed by a plane subtraction and line alignment. Three different spots (one in main text, two in Fig. S3) were measured on the sample to confirm uniformity and get a comprehensive view of the sample's features.

**Fluorescence correlation spectroscopy**. Fluorescence time traces were obtained by focusing a 640-nm laser line through a water immersion objective lens (×60 Plan Apo VC, N.A. 1.2, Nikon, Japan) at ~50 μm above the bottom of the glass-bottom 96-well plate (Grainer Bio-one, Frickenhausen, Germany). The measurement concentration was 1 μM Cy5-SLP in 4 mM of unlabeled SLPs and PBS (pH 7.4). After 5 min of preparatory work, 50 μl of sample was measured with confocal microscope (Nikon C1). Photon counting instrument (PicoHarp 300, PicoQuant) was used to record time trace by binning the photon counts in intervals of 600 s. Autocorrelation curves were fitted by using QuickFit 3.0[47] using three-dimensional diffusion with triplet as described below:

$$G(\tau) = \sum_{i=1}^{N} \alpha_i \left(1 + \frac{\tau}{\tau_{D_i}}\right)^{-1} \left(1 + \frac{\tau}{\gamma^2 \tau_{D_i}}\right)^{-\frac{1}{2}},$$

where $\tau$ represents lag time, $\tau_{D_i}$ is the diffusion time of the sample component, and $\gamma$ is the aspect ratio of the focal volume (~6 for common confocal microscope), $N$ is the number of discretization steps to sample the maximum entropy distribution (MEM), and $\alpha_i$ is the relative amplitude of the component. The MEM methodology works toward maximization of Shannon–Jaynes entropy (S), which is defined as

$$S = -\sum_{j=1}^{N} \rho_j \ln \rho_j,$$

where $\rho_j$ represents the probability of detecting a component in the confocal volume.

$$\rho_j = \frac{\alpha_j \tau_{D_j}}{\sum_{i=1}^{N} \alpha_i \tau_{D_i}}.$$

The diffusion coefficient $D$ is derived from:

$$D = \frac{w_{xy}^2}{4\tau_D},$$

where $w_{xy}$ is the lateral radius of the focal volume. $w_{xy}$ was calibrated with a solution of Alexa-647 ($D = 330$ μm$^2$/s at 25 °C), giving $w_{xy} \sim 300$ nm.

**ssNMR spectroscopy**. 1D CP[48] and scalar[49] ssNMR experiments to probe rigid and mobile populations, respectively, were acquired at 11.7 T magnetic field (500 MHz ($^1$H frequency) with 10 kHz magic angle spinning (MAS) and 280 K sample temperature. 2D $^{13}$C–$^{13}$C and 2D CαN for peptide assignments were also performed at similar conditions[37]. 2D Ca(N)H experiments for $^{15}$NT$_{1rho}$ relaxation studies were performed at 950 MHz magnetic field strength with 60 kHz. $^{13}$C was detected in the indirect dimension because of spectral overlap in the $^{15}$N dimension. The 2D $^{13}$C–$^{13}$C PARIS[37] experiment with the CPMAS CryoProbe[39] to measure intermolecular peptide contacts was performed at 600 MHz and 12 kHz MAS using 700 ms magnetization transfer and a $^1$H recoupling amplitude of 6 kHz.

**NMR structure determination**. An NMR structure of the SLP1 dimer was obtained using HADDOCK version 2.4[50] using default parameters. In total, we used 6 NMR chemical shift-derived[40] dihedral angle restraints and 60 intermolecular NMR distance restraints. The resulting dimer structure was very well defined and scored all within the same cluster.

**MD simulations**. Coarse-grained (CG) MD simulations were performed with the Martini force field version 2.2[51] and GROMACS 5.0.4[52]. Thirty-six SLPs were randomly immersed into a box with water beads to which 140 mM of NaCl was added (including neutralizing counterions). The systems were energy minimized and simulated for 3 μs (150,000,000 steps of 25 fs) using standard settings for nonbonded interactions in an NPT ensemble. Simulated systems were weakly coupled to a pressure bath at 1 bar ($\tau_p = 3$ ps) and coupled to a heat bath of 300 K temperature ($\tau_T = 1.0$ ps) using Berendsen algorithm[53]. Since Ile and Leu share the same CG beads in the Martini force field, SLP1–2 and SLP3–4 share the same CG models. Secondary structure of the SLPs were assigned as random coil with the martinize.py script. Random coil secondary structures were used in CG MD simulations to represent the initial steps of self-assembly. Polar P4 backbone beads were used to represent the N-terminal acetylate group. The final trajectories were transformed[54] to the atomistic coordinates and subject to the atomistic simulation.

Atomistic MD simulation were performed with the g53a6 force field[55] in GROMACS 5.0.4[52]. The fibril and oligomer models were first immersed into a box

of simple point charge (SPC) water[56] to which 140 mM of NaCl was added (including neutralizing counterions). The systems were energy minimized, then equilibrated for 100 ps in NVT ensemble at 300 K using V-rescale thermostat[57]. After that, 100 ps of NPT equilibration were performed at 1.0 bar using Parrinello–Rahman barostat[58]. Finally, the systems were simulated for 100 ns without restraints. The trajectories are available as Supplementary Data 1.

**SMD simulation**. SMD simulation were performed with the g53a6 force field[55] in GROMACS 4.5.5[52]. The equilibrated (100 ns) structures from the atomistic MD simulation were transferred to larger rectangular boxes of SPC water to which 140 mM of NaCl was added (including neutralizing counterions). The systems were energy minimized, then briefly equilibrated for 100 ps in NVT ensemble at 300 K, followed by 100 ps in an NPT ensemble at 1.0 bar. After that, 200 ps of NVE ensemble was performed with position-restrained peptides to optimize the charged side-chain orientations. Following the NVE equilibration, 1 SLP was pulled-out from the core of each structures along $x$-coordinate using a force constant of 1000 kJ/mol/nm$^2$ and pull-rate of 0.01 nm/ps. The other 35 SLPs were constrained by applying a harmonic force along the $x$-direction. The measurements of force and displacement of individual trajectories were saved every 10 fs. From these recorded trajectories, we derived the PMF profiles using Jarzisky's equality[41,42]. The trajectories are available in Supplementary Data 1.

**Reporting summary**. Further information on research design is available in the Nature Research Reporting Summary linked to this article.

## Data availability
The MD trajectories used in this study are available as Supplementary Data 1. Other related data that support the findings of this study are available from the corresponding authors upon reasonable request.

## Code availability
WALTZ and TANGO are freely accessible for academic and non-profit users at http://tango.crg.es/tango.jsp and http://waltz.switchlab.org/. QuickFit 3.0 is freely accessible for academic and non-profit users at https://github.com/jkriege2/QuickFit3.

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

## Acknowledgements

C.Y.J.L. acknowledges the support from the European Union (Horizion 2020 NANOMED Grant 676137). We thank Lione Willems (Wageningen University, The Netherlands) for her support in AFM and the Netherlands Center for Multiscale Catalytic Energy conversion (MCEC), an NWO Gravitation program funded by the Ministry of Education, Culture and Science of the government of The Netherlands, for the financial support with the AFM measurements; Javier Sastre Toraño (Utrecht University, The Netherlands) for his support in ESI-MS. Kevin Braeckmans (Ghent University, Belgium) for his advice on MEM analysis. M.W. acknowledges financial support (project numbers 723.014.003 and 711.018.001) from the Dutch Research Council (NWO). F.F. and F.G. acknowledge the support from the Italian Ministry of Health (Ricerca Corrente 2018–2020). The secondment of F.F. at Utrecht University was granted by the Erasmus Traineeship Program of University of Milano-Bicocca. We thank Professor Dr. Alexander Kros (Leiden University, The Netherlands) for critically reviewing this manuscript before submitting it for publication.

## Author contributions

C.Y.J.L., A.M.J.J.B., and M.W. contributed to the coarse-grained and atomistic molecular dynamics (MD) simulation. C.Y.J.L., F.F., and F.G. contributed to the steered MD simulation. C.Y.J.L., H.Z., and K.R. contributed to the FCS measurement and analysis. L.D.B.M. and R.d.V. contributed to the AFM measurement. D.W. contributed to the XRD measurement. J.v.d.D. contributed to the negative-staining TEM measurement. J.M.-S., B.V., M.W., A.H., B.P., and R.K. contributed to the ssNMR measurements. C.Y.J.L., W.H., M.W., and E.M. provided advice on the design of the whole experiments. C.Y.J.L., M.W., and E.M. designed the research concept, managed the project, and were the main contributors to the manuscript writing.

## Competing interests

The authors declare no competing interests.
