## [Peer Review File · Communications Chemistry]

Reviewers' comments:

Reviewer #1 (Remarks to the Author):

The manuscript by Chun Yin Jerry Lau et al. entitled "Control over the fibrillization yield by varying the oligomeric nucleation propensities of self-assembling peptides" reports the investigation of oligomeric nucleation propensities and nanofiber structure of self-assembling peptides using solid-state NMR, Fluorescence Correlation Spectroscopy (FCS), AFM, XRD, TEM and MD simulations. The authors show clearly that the difference of self-assembly manner with the four-type peptides. This study may be valuable for the biomedical applications on the rational design of self-assembling peptides. I recommend a major revision of this manuscript considering the following issues.

1.

The authors chemically synthesized four-type peptides (SLP-1, 2, 3, 4) by means of Fmoc SPPS method. Apart from that, C-terminal Cy5-ligated SLP peptides were prepared for FCS analysis. I wonder that this C-terminal modification sometimes affects the solubilization of the peptide and the self-assembly manner. Did you check it using another method (e.g. Dynamic light scattering DLS)?

2.

The authors recorded PARIS ^{13}C - ^{13}C spectra of self-assembled ^{13}C -labeled SLP-1. The spectra showed intra- and inter-molecular correlations. I'm not sure that there is ambiguity in an estimation of internuclear distance based on solid-state ^{13}C - ^{13}C NMR experiments such as PDSD and DARR. The authors made a model of simple anti-parallel dimer in Figure 4 (b). If there is ambiguity in the internuclear distance by PARIS, the authors should consider more complexed system. I think that it is necessary for the deep construction of a self-assembled manner to consider the patterns of in-register or out-of-register anti-parallel structure. How do you think about?

3.

From the FCS of SLP-1 (in Figure 2 c), the distribution of size can be confirmed. Is it deduced from different self-assembly? Does this distribution of SLP-1 affect the NMR results?

4.

The many reports with sequence- or stereochemical-dependent self-assembly of peptide have been published (Nature Chemistry (2010) 2, 1010-1011, Nanomedicine and nanobiotechnology (2013) 5, 582-612., Phys. Chem. Chem. Phys. (2019) 21, 10879-10883). I can't follow the novelty of this manuscript. Thus, the part of introduction must be improved.

Reviewer #2 (Remarks to the Author):

Overall, the topic of the manuscript is interesting and the authors have provided some meaningful results to support the novelty or significance of this study. Reasonable revisions are needed before being considered for acceptance.

Comments and suggestions:

1, the authors are suggested to compare with the other similar systems and discuss the novelty of this study

2, for future biomedical applications, biocompatibility tests are suggested to be performed

3, it's better for the authors to give the detailed yield difference since the title is about the yield

Reviewer #3 (Remarks to the Author):

I have carefully reviewed the article titled "Control over the fibrillization yield by varying the oligomeric 2 nucleation propensities of self-assembling peptides". Below is my assessment.

The Authors synthesized 4 surfactant-like peptides (SLPs) with small molecular variations :1- adding B-sheet propensity high (I) and low (L) amino acids; and 2- changing the number of negatively charged amino acid (E).

Using both MD simulations (atomistic, coarse-grained and steered) and some experimental techniques, they found that the nucleation propensity of the oligomeric intermediates and the thermodynamic stability of the fibril structures can be controlled by two parameters. In more detail, they found a correlation between the head group charge and the size of oligomers; the size of oligomers decreases with the increasing number of negative charges in the peptide head group.

Their findings could be useful understanding the importance of the effects of the molecular variations on fiber formation (in terms of both the thermodynamics properties and the final shape)

This is a clearly written article on an important topic. I recommend its publication.

In case, they might skip attention, I would like to point out to two minor points.

1: In line 60 "van der Waal's " should be " van der Waals".

2: The sentence starting in line 79 and ending in line 81 needs to be corrected.

A point-to-point response to the reviewers' comments:

We would like to thank the reviewers for their evaluation of our manuscript and their invaluable and constructive comments that has helped us to further improve the quality and clarity of our manuscript. Following is a list in which each query has been addressed point-to-point and there where appropriate changes have been made to the manuscript. We have highlighted these changes in yellow.

Reviewer #1 (Remarks to the Author):

Comments: The manuscript by Chun Yin Jerry Lau et al. entitled “Control over the fibrillization yield by varying the oligomeric nucleation propensities of self-assembling peptides” reports the investigation of oligomeric nucleation propensities and nanofiber structure of self-assembling peptides using solid-state NMR, Fluorescence Correlation Spectroscopy (FCS), AFM, XRD, TEM and MD simulations. The authors show clearly that the difference of self-assembly manner with the four-type peptides. This study may be valuable for the biomedical applications on the rational design of self-assembling peptides. I recommend a major revision of this manuscript considering the following issues.

Answer: Thank you very much for the kind remark. Below you will find our response.

1.

The authors chemically synthesized four-type peptides (SLP-1, 2, 3 ,4) by means of Fmoc SPPS method. Apart from that, C-terminal Cy5-ligated SLP peptides were prepared for FCS analysis. I wonder that this C-terminal modification sometimes affects the solubilization of the peptide and the self-assembly manner. Did you check it using another method (e.g. Dynamic light scattering DLS)?

Answer: Thank you for raising this very legitimate concern. We have indeed taken this into account in our initial molecular design. To minimize the influence of the fluorophore towards the self-assembly behavior, we have coupled the fluorophore to the headgroup domain, away from the tail domain, which is driving the self-assembly. Furthermore, instead of labelling peptides at one of the headgroup residues, we have also labelled the SLPs at the extended lysine residues. In this way we can preserve the amount of charges present in the SLPs. This fluorophore labelling strategy has been employed before and was shown not to disturb the self-assembly manner of similar SLPs as determined by super-resolution microscopy (Biomacromolecules 2017, 18, 11, 3481–3491, see figure below for the STORM image reported). Furthermore, using the preparation protocol as described in the method section (line 276-279, page 6), we are able to prepare the SLP-Cy5 assemblies at a concentration of 4mM without encountering any solubilization issues. Comparison of size distribution between the Cy5 vs nonlabelled peptides using DLS was hampered by the excitation of the Cy5 probe with the DLS laser. Besides, the SLP assemblies have a rather high polydispersity in size, making it a less suitable samples for DLS measurements.

2.

The authors recorded PARIS ¹³C-¹³C spectra of self-assembled ¹³C-labeled SLP-1. The spectra showed intra- and inter-molecular correlations. I'm not sure that there is ambiguity in an estimation of internuclear distance based on solid-state ¹³C-¹³C NMR experiments such as PDS and DARR. The authors made a model of simple anti-parallel dimer in Figure 4 (b). If there is ambiguity in the internuclear distance by PARIS, the authors should consider more complexed system. I think that it is necessary for the deep construction of a self-assembled manner to consider the patterns of in-register or out-of-register anti-parallel structure. How do you think about?

Answer: We are grateful for the referee's comment. Distances derived from solid-state NMR ¹³C-¹³C spin diffusion experiments are usually applied as restraints with a maximal distance threshold of 6-8 Å and the sum of the van der Waals radii as lower distance threshold. The sheer number of restraints then determines the precision of the structure determination. For our peptide dimer, we succeeded to collect more than 50 intermolecular distance restraints and all calculated dimers scored an out-of-register dimer in the same HADDOCK cluster, indicating a high accuracy and confidence in the structure determination. What is more, an in-register assembly is not in line with the obtained NMR distance pattern. With the large number of distance restraints and the high accuracy of our structure determination, we do not see cause to doubt the NMR dimer structure.

3.

From the FCS of SLP-1 (in Figure 2 c), the distribution of size can be confirmed. Is it deduced from different self-assembly? Does this distribution of SLP-1 affect the NMR results?

Answer: Thank you for raising this point. As described in line 105-108, page 3 of the revised manuscript, we have employed the maximum entropy method to fit the FCS data to derive the size distribution. Regarding the bimodal size distribution observed in SLP1, as we stated in line 114-116, page 3 of the revised manuscript, we are unable to distinguish the identity of the assemblies in the larger sized population between larger oligomers and the early nucleated structures. However, we can validate our hypothesis that the number of headgroup charges are critical determinants in regulating the size of the oligomers. We have designed tail group of SLPs consisting of two chemically anisotropic inter-side chain interfaces (Figure S3), which helps directing the cross-β molecular arrangement. Indeed, we have recorded a comparable dipolar spectrum between SLP1-3, indicating that the distribution of SLP1 doesn't affect the NMR results in structural determination.

4.

The many reports with sequence- or stereochemical-dependent self-assembly of peptide have been published (Nature Chemistry (2010) 2, 1010–1011, Nanomedicine and nanobiotechnology (2013) 5, 582-612., Phys. Chem. Chem. Phys. (2019) 21, 10879-10883). I can't follow the novelty of this manuscript. Thus, the part of introduction must be improved.

Answer: We regret that the introduction of our manuscript was written in a way that the novelty was not properly conveyed, and we therefore thank the reviewer for pointing this out. Our manuscript is different from the many accounts on peptide self-assembly in the fact that we highlight the importance of the intermediate steps in self-assembly, which critically determines the yield of the supramolecular

assembly process. Most articles on peptide self-assembly only focused on the final stage of the self-assembly process, assuming a one-step assembly process from monomeric peptide to fibrils. Therefore, the role of the intermediate states is often ignored. In contrast, our manuscript has described how molecular variation in the peptide sequence can moderate the properties of the oligomeric intermediates, which determines the yield of the final assembled products.

We have adjusted the introduction to better describe the above-mentioned novelty of our findings as follows.

In line 46-50, page 1-2:

“In search for the link between the peptide monomers and the final assembly states, a number of previous reports on self-assembling peptides have studied the effect of changing molecular properties of the peptide monomer (e.g. sequences’ residues or stereochemistry^{8,9}) towards the final assembled products. Many studies have treated the peptide self-assembly as a spontaneous thermodynamic process^{8,9,14}. Therefore, the linear correlation between the properties of peptide monomers and the final assembled structures are often described in these studies.”

In line 56-58, page 2:

“However, despite progress in the mechanistic understanding, the inter-relationship between the molecular properties of the peptide monomers, the oligomeric intermediates and the overall yield of the supramolecular assembly process remains largely elusive²⁶. Here, we used surfactant-like peptides (SLPs) to show that the yield of peptide fibrillization are controlled by the properties of the oligomeric intermediates, which can be moderated by subtle molecular variations in the peptide sequences.”

Reviewer #2 (Remarks to the Author):

Overall, the topic of the manuscript is interesting and the authors have provided some meaningful results to support the novelty or significance of this study. Reasonable revisions are needed before being considered for acceptance.

Answer: We very much appreciate the reviewer’s effort in evaluating our manuscript. Below we have addressed all questions raised by the reviewer point-by-point.

Comments and suggestions:

1, the authors are suggested to compare with the other similar systems and discuss the novelty of this study

Answer: We thank the reviewer for the advice. This point has been addressed in the reply to comment 4 of reviewer 1.

2,for future biomedical applications, biocompaibiity tests are suggested to be performed

Answer: We thank the reviewer for the advice. To validate the biocompatibility of the SLP assemblies, we have performed a viability assay (MTS) for incubating the SLP assemblies with human skin carcinoma (A431) cell line for 24 hours (see below). The results have been added to the SI as S9.

3, it's better for the authors to give the detailed yield difference since the title is about the yield

Answer: We thank the reviewer for the advice. As the intensity of the dipolar spectra is correlative to the amount of cross- β structure present in the system, by observing the spectra intensity difference, we can derive a relative fibrillization yield ranking between SLP1-3. We have modified the paragraph in line 143-148 of the revised manuscript as follow to detail the fibrillization yield difference.

In line 143-148, page 3-4

“A closer inspection into the population profiles of SLP1-3 revealed noticeable difference in the dipolar and scalar signals between SLP1-3. The intensity of the dipolar spectra correlates with the amount of rigid cross- β structure present in the system. We observe that the intensities of dipolar signal of SLP1 is larger than SLP2 (normalization scaling factor of 10, Figure 3b) and SLP3 (normalization scaling factor of 25, Figure 3b), which indicate that the relative fibrillization yield of SLP1>SLP2>SLP3.”

Reviewer #3 (Remarks to the Author):

I have carefully reviewed the article titled “Control over the fibrillization yield by varying the oligomeric 2 nucleation propensities of self-assembling peptides”. Below is my assessment.

The Authors synthesized 4 surfactant-like peptides (SLPs) with small molecular variations :1-adding B-sheet propensity high (I) and low (L) amino acids; and 2- changing the number of negatively charged amino acid (E).

Using both MD simulations (atomistic, coarse-grained and steered) and some experimental techniques, they found that the nucleation propensity of the oligomeric intermediates and the thermodynamic stability of the fibril structures can be controlled by two parameters. In more detail, they found a

correlation between the head group charge and the size of oligomers; the size of oligomers decreases with the increasing number of negative charges in the peptide head group.

Their findings could be useful understanding the importance of the effects of the molecular variations on fiber formation (in terms of both the thermodynamics properties and the final shape)

This is a clearly written article on an important topic. I recommend its publication.

In case, they might skip attention, I would like to point out to two minor points.

1: In line 60 “van der Waal’s “ should be “ van der Waals”.

2: The sentence starting in line 79 and ending in line 81 needs to be corrected.

Answer: Thank you very much for the generous comments and useful advices. The modified sentences can be found at line 61 and 81-83 in the revised manuscript and are shown in below.

In line 61, page 2:

“...hydrophobic tails that interact via van der Waals’ forces (side chain) and hydrogen bonds (backbone)..”

In line 81-83, page 2:

“The length of the tail group was chosen as N-terminal acetylated 7-mer peptide. An extra amino acid and a N-terminal acetyl cap was added to the 6-mer peptide sequence to minimize the influence of flanking effects by the charged residues.”

REVIEWERS' COMMENTS:

Reviewer #1 (Remarks to the Author):

The revised manuscript and the author's response were checked.
I'm satisfied with the author's response.
I recommend strongly the acceptance of the revised manuscript to Communications Chemistry.

Reviewer #2 (Remarks to the Author):

The revised manuscript is OK for acceptance now